# Long-term Exposure to Air Pollutants and Alzheimer's Disease Dementia Prevalence Across the Contiguous United States: An Explainable Machine Learning Analysis

Oliver Aschenbrenner
*Department of Computer Science*
*College of Charleston*
Charleston, South Carolina, USA
aschenbrennerog@g.cofc.edu

Navid Hashemi Tonekaboni
*Department of Computer Science*
*College of Charleston*
Charleston, South Carolina, USA
hashemin@cofc.edu

Mackenzie Kramer
*College of Computing Science*
*Clemson Univeristy*
Clemson, South Carolina, USA
mjkrame@g.clemson.edu

Abe Mollalo
*Deptartment of Public Health*
*Medical University of SC*
Charleston, South Carolina, USA
mollalo@musc.edu

*Abstract— A growing number of studies have examined the relationship between environmental factors and Alzheimer's Disease (AD) dementia prevalence. However, exploration into long-term exposure to air pollutants at the county level across the United States using spatial machine learning has been insufficiently studied. We compiled long-term data for six air pollutants ($PM_{2.5}$, $PM_{10}$, $NO_2$, CO, $O_3$, and $SO_2$) from 1999 to 2020 to evaluate their relationship with AD dementia prevalence using global Random Forest, global XGBoost, geographically weighted random forest (GWRF), and local XGBoost models. These models were evaluated with several metrics (i.e. $R^2$, RMSE, and AIC). Moreover, Gini feature importance and SHAP values were used to assess the relative contribution of each pollutant and interpret model outputs. The GWRF model outperformed other local and global models, with an $R^2$ value of 54.38%, with the best fit observed in the Northeast and West Coast regions. Findings from Gini feature importance showed $PM_{10}$ as the most influential predictor, followed by $NO_2$, $O_3$, and $PM_{2.5}$. In addition, $PM_{10}$ emerged as the primary variable in 25.31% of counties (n=786), while $SO_2$ and CO had a smaller role. Our results suggest that, among air pollutants, $PM_{10}$ may play a more significant role in AD dementia prevalence than previously recognized, especially in urban areas.*

*Keywords— Air pollutants, Alzheimer's disease dementia, Geographically weighted random forest, Spatial machine learning*

## I. INTRODUCTION

Most research on Alzheimer's disease (AD) dementia has traditionally focused on individual-level factors, but growing evidence shows that broader social and environmental determinants also play a significant role in developing AD dementia risk [1]. The socio-economic burden of environmental exposures on AD dementia is substantial. In the US, air pollution is linked to a loss of independence in approximately 730,000 older adults each year, with related costs nearing $11.7 billion [2]. Exposures to noise, air pollution, and urban heat have been shown to contribute to cognitive decline over time. For example, long-term exposure to traffic-related noise is associated with lower cognitive performance in older adults [3], while neighborhood stressors and environmental degradation have been related with more severe neuropsychiatric symptoms among people living with dementia [4]. Moreover, limited access to green space has been associated with cognitive decline [5].

Recognizing the impacts of environmental determinants, researchers have increasingly focused on air pollution as a widespread and modifiable risk factor for AD dementia. A systematic review and meta-analysis by Tsai et al. (2019) found that fine particulate matter ($PM_{2.5}$) was significantly and positively associated with dementia, with a pooled hazard ratio of 3.26, indicating more than triple the risk among those exposed to higher levels of $PM_{2.5}$ levels [6]. More recently, Tang et al. (2023) conducted a meta-analysis and demonstrated a significant increase in dementia risk with higher levels of air pollution [7]. Their analysis found that dementia risk increased with exposures to $PM_{2.5}$, nitrogen dioxide, and carbon monoxide, among others [8]. Experimental studies also support these findings. Rahman et al. (2020) found that airborne pollutants increase amyloid β peptide and tau phosphorylation which can contribute to the development of amyloid plaques, a key feature of AD dementia [9].

Among commonly used machine learning (ML) models, Random Forest (RF) and Extreme Gradient Boosting (XGBoost) are particularly effective for environmental health research due to their ability to model complex interactions and rank feature importance [11,12]. While these models can capture complex relationships, they generally assume spatial stationarity. However, environmental exposures and their health impacts often vary geographically due to differences in pollution sources, infrastructure, and sociodemographic characteristics. To address this limitation, researchers have increasingly adopted local models—such as Geographically Weighted Random Forest (GWRF) or local XGBoost—which allow the relationship between predictors and outcomes to vary across geographic areas [11, 12]. For instance, a study by Mollalo et al. (2025) used a GWRF model to examine county-level AD dementia prevalence across the US. Their findings revealed substantial regional variations in the influence of environmental and social risk factors on AD dementia prevalence [10].

In this study, we hypothesize that long-term exposure to air pollution significantly contributes to AD dementia prevalence

across the US and the relationship varies by geographic region. Specifically, this study aims to address the following questions:

(1) How do global vs. local ML models perform in predicting county-level AD dementia prevalence?

(2) How does the air pollution–AD dementia relationship vary spatially at the county level across the contiguous US?

(3) What is the relative importance of air pollutants in predicting AD dementia prevalence across the contiguous US?

The findings from this study aim to offer place-based insights into the role of air pollutants in the burden of AD dementia across the contiguous US, helping to inform more targeted and refined public health policies.

## II. METHODS

### A. Data Collection and Preparation

County-level estimates of AD dementia prevalence (n=3,142 counties) were obtained from Dhana et al. (2023) [14]. This study used data from the Chicago Health and Aging Project (CHAP), a large, population-based cohort of more than 10,000 adults aged 65 and older residing in Chicago, which included extensive neuropsychological testing and demographic information. To estimate the probability of AD dementia, Dhana et al. applied a generalized additive quasibinomial regression model adjusted for age, sex, race/ethnicity, and education. This model was then applied to 2020 bridged-race postcensal population estimates from the National Center for Health Statistics, stratified by demographic group, to produce demographically adjusted prevalence estimates for every US county. These spatially comprehensive estimates served as the foundation for our analyses.

Estimates for air pollution levels were obtained from the Center for Air, Climate, and Energy Solutions, which provides high-resolution exposure data at various spatial resolutions across the contiguous US based on a national land use regression model [15]. Air pollutant estimates are derived by integrating satellite remote sensing, ground-based monitoring data from the US Environmental Protection Agency, land use characteristics, and meteorological information [15, 16]. In this study, we used annual county-level estimates for six pollutants: $PM_{2.5}$ ($\mu g/m^3$), coarse particulate matter ($PM_{10}$, in $\mu g/m^3$), $NO_2$ (parts per billion [ppb]), CO (ppb), ozone ($O_3$, in ppb), and sulfur dioxide ($SO_2$, in ppb). To capture long-term exposure, the median concentration for each pollutant was calculated across the years 1999 to 2020.

### B. Global Models

After preparing the data, two global ML models were implemented—RF and XGBoost—to examine the relationship between AD dementia prevalence and the selected air pollutants.

### C. Local Models

While global models provide useful insights in understanding broad patterns, they overlook spatial non-stationarity. Local models are better suited to address spatial heterogeneity driven by regional variations in pollution sources, population characteristics, infrastructure, and social determinants of health [20]. Accordingly, this study employed two local ML models—GWRF and Local XGBoost—to allow the relationships to vary geographically and capture localized effects. Details on each local modeling approach are provided below.

1) Geographically Weighted Random Forest: GWRF models combine the strengths of RF with the spatial adaptability of Geographically Weighted Regression [21]. By training the decision trees locally, GWRF can capture location-specific relationships between air pollutants and AD dementia prevalence [21]. This is particularly important as environmental exposures and their health impacts are not uniform across space. GWRF, like RF models, can handle complex datasets, identify non-linear relationships, and is inherently robust to overfitting due to the use of a random sample of the data and a random selection of the input features for each tree [10]. The ability to obtain feature importance scores is also a benefit of GWRF models that may be masked in global models.

2) Local XGBoost: Another local model used in this study was the local XGBoost, which builds separate models for different geographic areas. The main benefit of using local XGBoost is that it combines the strengths of gradient boosting—such as strong predictive performance and the ability to model complex relationships—with the flexibility to adapt to local spatial contexts, improving accuracy across regions [22]. This approach helps capture region-specific patterns and may better reflect how environmental factors vary across communities, particularly where pollution sources and population vulnerabilities differ [22]. The model can improve prediction accuracy by sequentially learning from past errors and includes techniques such as regularization and tree pruning to reduce overfitting—ensuring the model generalizes well to new data [18, 22].

### D. Model Settings

To train, tune and evaluate the ML models, the full dataset was randomly partitioned into three subsets: 70% for training, 15% for validation, and 15% for testing. Model building was conducted on training dataset, while hyperparameter tuning was carried out on the validation set to reduce overfitting and improve generalizability. The final test set was used to evaluate out-of-sample performance, providing an unbiased estimate of how well each model generalized to unseen data. Out-of-bag (OOB) error estimates were generated for RF and GWRF models to provide an internal measure of accuracy without requiring a separate validation set. For the global XGBoost model, which lacks native OOB estimates, an OOB estimate was generated using 5-fold cross-validation on the training dataset. Leave-one-out cross-validation was implemented for the local XGBoost model. For each training observation, a local XGBoost model was trained using neighboring observations, excluding the observation itself. Then, the model predicted the excluded point, repeated for all training locations. These predictions were compared to actual values to compute $R^2$ and RMSE scores as OOB error estimates [17].

A grid search approach was used for hyperparameter selection, optimized through five-fold cross-validation using $R^2$ as the primary performance metric [23]. For RF, the hyperparameter search included the number of decision trees, maximum tree depth, number of features considered at each

split, and minimum sample thresholds for both leaf nodes and splits [24]. In the case of GWRF, tuning focused on the number of trees and tree depth, consistent with its adaptation for local modeling [24]. The XGBoost models—both global and local—were tuned across parameters including number of trees, maximum depth, learning rate, subsample ratio (i.e., fraction of observations used per tree), and column sampling ratio (colsample_bytree) [25]. Sensitivity analyses were conducted to assess the impact of different neighborhood sizes (k= 5 to 150) and kernel types (fixed vs. adaptive). The number of neighbors (k=92) used for local model fitting was determined through these analyses and using Golden Search optimization, with a fixed kernel providing the best balance of accuracy and spatial stability. These sensitivity analyses confirmed that the identified pollutant importance rankings and spatial patterns remained stable (TABLE I).

TABLE I.        TUNED HYPERPARAMETER VALUES FOR MODELS

| No. | Hyperparameter | RF | GWRF | Global XGBoost | Local XGBoost |
|---|---|---|---|---|---|
| 1 | n_estimators | 300 | 300 | 300 | 300 |
| 2 | max_depth | 30 | 30 | - | - |
| 3 | min_samples_leaf | 1 | 1 | - | - |
| 4 | min_samples_split | 2 | 2 | - | - |
| 5 | learning_rate | - | - | 0.05 | 0.05 |
| 6 | subsample ratio | - | - | 0.8 | 0.8 |
| 7 | colsample_bytree | - | - | 0.8 | 0.8 |

### E. Model Evaluation and Interpretability

Model performance was assessed using several evaluation metrics on the test set. These included the coefficient of determination ($R^2$), root mean squared error (RMSE), and Akaike Information Criterion (AIC). AIC was computed using the residual sum of squares, an estimate of residual variance (approximated via RMSE), and the number of predictors, offering a measure of model fit that also penalizes complexity [26].

To enhance model interpretability, feature importance measures were used to quantify the role of each predictor on AD dementia prevalence. Gini importance was calculated for both RF and GWRF models [17], and gain-based importance was applied to XGBoost models [19]. To further interpret and visualize these relationships, SHAP values were used to illustrate the direction and magnitude of each feature's impact on model predictions.

In addition to global performance metrics, spatial diagnostics were used to evaluate regional variation in model fit. For GWRF, local $R^2$ values were computed to assess the strength of model fit across counties [9]. To assess spatial clustering of residuals at the local level, Local Moran's I was calculated and mapped to visualize areas with high or low residual similarity and identify potential spatial clustering in model residuals.

All model development was performed in Python using libraries such as GeoPandas, NumPy, Pandas, scikit-learn, statsmodels, SHAP, esda.moran, libpysal, XGBoost, and PySAL. Spatial visualization and mapping of AD dementia prevalence and top-ranked predictors were performed using Matplotlib and Geopandas.

## III. RESULTS

### A. Descriptive Statistics

Preliminary statistics showed that the AD dementia prevalence across the contiguous US ranged from 5.6% in Loving, Texas to 18.4% in Presidio, Texas. The mean prevalence was 11.2%, with a median of 10.9% and a standard deviation of 1.4. Descriptive statistics for air pollutant concentrations are summarized in TABLE II. Each pollutant exhibited distinct spatial patterns and concentration extremes. $PM_{2.5}$ concentrations were highest in central California and the Southeast part of the US, while the lowest concentrations were observed in the Midwest and Southwest. $PM_{10}$ concentrations were highest in southern California and the Midwest, with lowest levels in the Northeast. $NO_2$ had a higher concentration in southern California and near New York City, while it showed the lowest concentrations in Mountain West and Great Plains. $SO_2$ concentrations peaked in the Ohio River Valley region, while lowest concentrations were towards the Western US. $O_3$ concentrations were highest in the southwest, while the lowest concentrations were observed in the Northeast and Southeast. CO concentrations were elevated in the Southwest, while lower concentrations were toward the East Coast. Fig. 1 depicts the geospatial distribution of air pollutants.

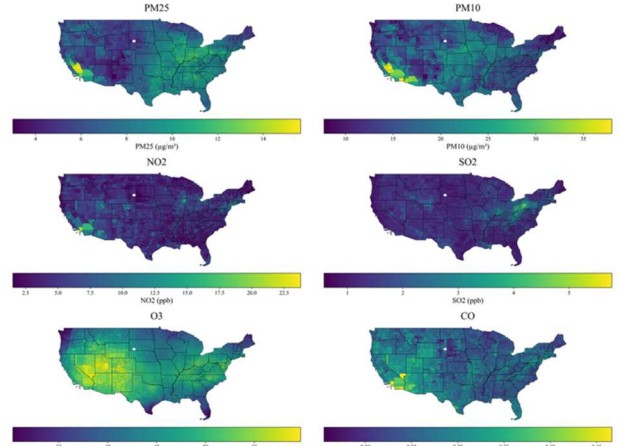

Fig. 1. $PM_{2.5}$, $PM_{10}$, $NO_2$, $SO_2$, $O_3$, and CO concentrations across the contiguous US.

TABLE II.        DESCRIPTIVE STATISTICS FOR EXPLANATORY VARIABLES

| No. | Pollutant | Minimum | Maximum | Mean | Median | Standard Deviation |
|---|---|---|---|---|---|---|
| 1 | $PM_{2.5}$ | 3.00 | 15.64 | 8.65 | 8.89 | 1.83 |
| 2 | $PM_{10}$ | 7.74 | 37.99 | 17.58 | 17.31 | 3.75 |
| 3 | $NO_2$ | 1.50 | 23.78 | 4.90 | 4.51 | 2.07 |
| 4 | $SO_2$ | 0.58 | 5.77 | 1.56 | 1.39 | 0.54 |
| 5 | $O_3$ | 30.31 | 59.72 | 46.32 | 47.04 | 4.43 |
| 6 | CO | 0.16 | 0.47 | 0.26 | 0.26 | 0.03 |

### B. Variable Selection

For the six initial features ($PM_{2.5}$, $PM_{10}$, $NO_2$, $SO_2$, CO, and $O_3$), the correlation coefficients ranged from 0.03 (between $PM_{10}$ and $SO_2$) to 0.64 (between CO and $NO_2$). Overall, there were weak to moderate correlations between the air pollutants, but no extreme correlations ($|r|<0.7$) that would necessitate the removal of an air pollutant. The variance inflation factors (VIFs) ranged from 1.21 ($O_3$) to 2.22 ($NO_2$), supporting the

inclusion of these variables in the modeling process. Pollution interaction terms (PM$_{2.5}$ x PM$_{10}$, SO$_2$ x NO$_2$, etc.) were initially considered as additional features. However, the resulting VIFs were extremely large, indicating a high degree of multicollinearity, and thus were excluded from further analysis.

## C. Model Performance Comparisons

Among the global models, RF could explain 40% of variations of AD dementia prevalence, compared to 36% in XGBoost. RF also exhibited lower error rates and a slightly lower AIC than global XGBoost, indicating that RF outperformed XGBoost overall (TABLE II). However, local Models outperformed global models. GWRF slightly outperformed Local XGBoost in model fit, explaining 54% of the variance in AD dementia prevalence compared to 52% by Local XGBoost. Although Local XGBoost had the lowest AIC, suggesting better generalization, GWRF demonstrated superior predictive performance with a slightly higher R$^2$ and substantially lower RMSE. Given that R$^2$ = 0.54 is considered a strong model fit in spatial epidemiological studies, where values above 0.4 are often interpreted as good model fit [27], GWRF was selected for further analysis based on its overall performance. TABLE III summarizes the model performance metrics.

TABLE III.    MODEL PERFORMANCE METRICS FOR GLOBAL VS. LOCAL MODELS IN AD DEMENTIA PREVALENCE

|  | *Model* | *R$^2$* | *RMSE* | *AIC* |
|---|---|---|---|---|
| **Global** | RF | 0.40 | 1.21 | 1511.36 |
|  | XGBoost | 0.36 | 1.25 | 1534.21 |
| **Local** | GWRF | 0.54 | 1.05 | 1389.95 |
|  | Local XGBoost | 0.52 | 1.94 | 629.15 |

## D. Geospatial Distribution of Primary Variables

Feature importance for the GWRF model was calculated using Gini importance and found that PM$_{10}$ had the highest feature importance (0.181), followed by NO$_2$ (0.178), O$_3$ (0.176), PM$_{2.5}$ (0.166), CO (0.157), and SO$_2$ (0.143). PM$_{10}$ emerged as the primary variable in 25.31% of counties (n=786). PM$_{10}$ was the primary variable in most Northern states such as Montana, North Dakota, Idaho, and Wyoming. O$_3$ followed closely, being the primary variable in 24.24% of counties (n=753), primarily in Southern counties in Georgia, Alabama, and Arkansas. NO$_2$ was the primary variable in 20.35% of counties (n=632), predominantly in Northeast and New England counties in states such as New York, Vermont, and Maine. PM$_{2.5}$ was the primary variable in 13.43% of counties (n=417), predominantly in Western US counties in California, Nevada, and Colorado. SO$_2$ was the primary variable in 9.14% of counties (n=284), scattered throughout the US, with larger clusters in Texas, North Carolina, and Virginia. CO was the primary variable in 7.53% of counties (n=234), and was also scattered throughout, with spatial clustering in Texas, Minnesota, and Iowa. Fig. 2 depicts the geographic distribution of the primary variables across the contiguous US.

In addition to Gini importance, SHAP values were used to interpret contributions of individual air pollutants to AD dementia prevalence predictions. Among the pollutants, PM$_{10}$ had the highest SHAP values, indicating it contributed the most to

variation in AD dementia prevalence predictions. O$_3$, NO$_2$, and PM$_{2.5}$ followed, displaying intermediate SHAP value distributions. SO$_2$ and CO displayed lower SHAP values overall, suggesting a smaller role in influencing GWRF model's predictions. Also, the range and density of SHAP values for PM$_{10}$ was broader than the other air pollutants, showing more variability in its contribution. Fig. 3 shows SHAP values for air pollutants from the GWRF model highlighting the relative importance and the direction of influence of key predictors on AD dementia prevalence.

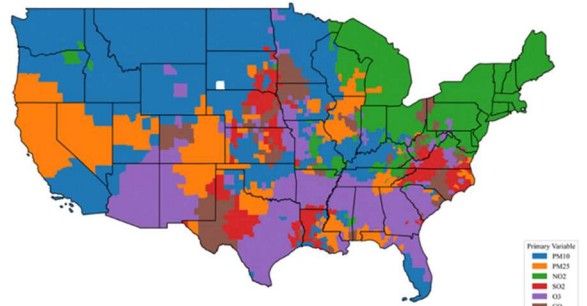

Fig. 2. Primary variables associated with AD dementia prevalence determined by GWRF across the contiguous US.

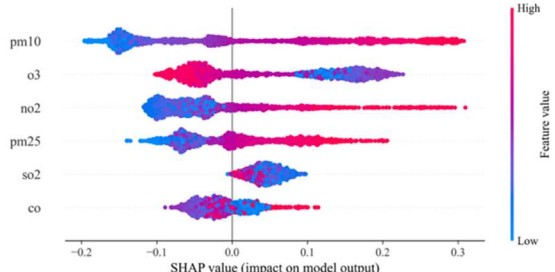

Fig. 3. SHAP values for GWRF model.

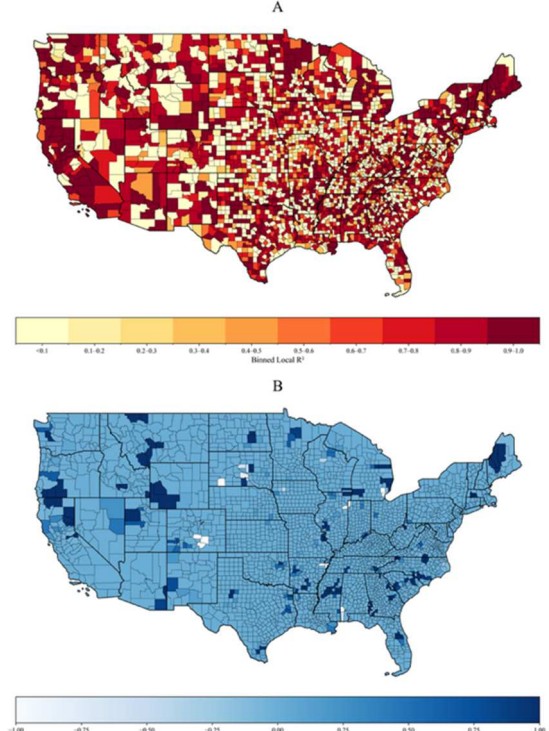

Fig. 4. Variations of (A) Local R$^2$ values and (B) Local Moran's I residuals for GWRF Model

## E. Geographical Variations in Model Fit

The geospatial distribution of model fit varied substantially by location. The GWRF model showed a better fit ($R^2 \approx 0.7$) in the Northeast and along the West Coast, especially in Pennsylvania, New York, New Jersey, and California. Moderate model fit ($R^2 \approx 0.3$) was observed in the upper Midwest and Great Lakes area, central Texas, and Northwestern portion of the US. The worst model performance ($R^2 < 0.1$) was observed across the Central US, with spatial clustering especially in Utah, Arkansas, and Minnesota. Fig. 4A shows the spatial distribution of local $R^2$ values, binned in 10% increments. Moreover, local Moran's I showed that the residuals of the GWRF exhibit mainly random patterns with a few hotspots or cold spots indicating areas of overprediction or underprediction (Fig. 4B).

## IV. DISCUSSION

This study explored the relationship between long-term exposure to air pollution and AD dementia prevalence across the contiguous US. RF and XGBoost were applied at global and local levels to assess the model performance and key predictors. GWRF showed the best fit and lowest error and was used for further analysis. Model fit was highest in urban areas, likely due to elevated pollutant levels enhancing predictive power. The GWRF model revealed that $PM_{10}$ had the highest overall feature importance followed by $NO_2$, $O_3$, and $PM_{2.5}$. $PM_{10}$ was the most common primary variable in over 25% of counties, with $O_3$ and $NO_2$ following. This suggest that higher concentrations of particulate matter–particularly $PM_{10}$–are associated with higher AD dementia prevalence, underscoring the potential effectiveness of localized air quality interventions aimed at reducing $PM_{10}$ concentrations to mitigate AD dementia burden.

Although air pollution levels have improved in many regions since the 1990s, our exposure estimates were calculated using long-term average pollutant concentrations from 1999 to 2020. This 22-year window reflects chronic exposure, which is more biologically relevant for neurodegenerative diseases with long latency periods, such as AD dementia. Prior studies have also used long-term historical exposure windows to examine associations with cognitive outcomes [46, 47]. Growing evidence suggests that early- or mid-life exposure to air pollution can trigger long-term neuroinflammatory processes linked to AD, making chronic exposure relevant even as pollution levels decline [48]. The spatial patterns of the observed primary variables align with notable sources of pollution and environmental characteristics. $PM_{10}$ and $PM_{2.5}$ had the largest impacts in the Northwest and West Coast, potentially due to the presence of wildfires in the area. Wildfires are a growing source of both $PM_{2.5}$ and $PM_{10}$, and since with wildfires they cannot always be fully contained, the increase in particulate matter has likely led to higher impact in these regions [28]. Higher impact of $NO_2$ in the Northeast could be due to elevated traffic emissions from higher traffic density. $NO_2$ is a key pollutant produced by vehicle emissions, and its high concentration in densely populated urban areas like New York City have been detailed in a previous study [29]. $O_3$ had the greatest impact in Southern counties, which can be attributed to warmer climate, abundant sunlight, and high industrial activities with the burning of fossil fuels. $O_3$ concentrations in this region have been recorded as increasing, reaching unusually high levels [30].

Additionally, $SO_2$ and CO's importance had spatial clustering throughout the South and Midwest. In these areas, the combustion of fossil fuels is a major source of $SO_2$ and CO. Power plants and industrial facilities are key contributors to $SO_2$ emissions, while CO is produced from inefficient combustion [31, 32].

We found noticeable regional differences in model fit, with the best model fit concentrated in the Northeast, upper Midwest, and parts of the West Coast. Contrastingly, spatial clusters of lower $R^2$ values were found, especially in the Great Plains and Central US, indicating the lower role of air pollutants in these areas. These patterns suggest that other region-specific factors may be missing from the model, or likely reflect regional differences in data quality, population density, reporting practices, and pollutant variability. In areas with poor performance, results should be interpreted cautiously, but overall, the GWRF model captures meaningful spatial patterns in the pollutant-AD dementia relationship. While the primary aim of this study was to focus exclusively on air pollutants, future modeling may benefit from including other locally relevant covariates, such as social determinants of health or healthcare access, or using other alternative local modeling such as spatial Bayesian approach to better capture the complexity of spatial variation in AD dementia prevalence.

The superior performance of the GWRF model underscores the value of incorporating spatial dependencies into ML to capture local relationships between air pollutants and AD dementia prevalence. Unlike global models that assume uniformity, GWRF reflects local variations, offering a more nuanced view at the county level. This aligns with prior work: Lotfata et al. (2023) reported that GWRF outperformed RF in modeling asthma prevalence [33], and Grekousis et al. (2022) showed similar results for COVID-19 mortality based on demographics [21]. While associations between air pollutants and AD dementia have been explored, few studies apply spatial ML approaches.

A key contribution of this study is the identification of $PM_{10}$ as the air pollutant most strongly associated with AD dementia prevalence. While $PM_{2.5}$ has been extensively detailed in existing literature due to its ability to penetrate deeper into the respiratory system, $PM_{10}$ may have more spatially variable effects from sources like roads, transportation, and construction sites [34]. $PM_{10}$ emerged as the top predictor probably due to its greater spatial variability at the county level and stronger localized signals from the previously listed sources. In contrast, $PM_{2.5}$ may have shown more spatially uniform effects across regions, lowering its relative importance in the GWRF model. Although $PM_{10}$ has been included in some previous systematic reviews, such as that by Meo et al. (2024), its importance has often been underemphasized relative to $PM_{2.5}$ [35]. As our study was conducted at the county level, discrepancies with individual-level studies may reflect differences in spatial scale and exposure assessment. However, emerging evidence, including findings by Ning et al. (2023), imply that $PM_{10}$ exposure may contribute to an elevated risk of AD dementia [36]. Our findings suggest the need for further investigation into the potential neurological effects of $PM_{10}$ exposure. To do so, we plan to (1) conduct multilevel analyses combining individual and contextual level data, (2) use finer-resolution estimates, and

(3) assess effect modification by rural-urban status and social vulnerability. We propose several public health interventions that may mitigate AD dementia related to $PM_{10}$ exposure: (1) strengthen air quality regulations targeting PM10 sources (e.g., traffic, construction, industry); (2) promote urban greening to reduce pollutants; (3) expand clean public transit to lower emissions; and (4) increase awareness and cognitive screening in high-exposure communities.

$NO_2$ emerged as the second most important predictor in our GWRF model, somewhat aligning with existing literature. $NO_2$, a pollutant that primarily gets in the air from the burning of fuel, has been shown to trigger neuroinflammation, thus is connected to neurodegenerative diseases [36, 37]. Zhang et al. (2021) identified a strong positive association between $NO_2$ exposure and AD dementia emergency room visits in a nationwide study of over 7.5 million cases [38]. Our findings support this relationship, especially in densely populated counties with higher $NO_2$ concentrations. Similarly, Mork et al. (2023) found that long-term exposure to $NO_2$ was associated with accelerated risk of AD-related hospitalization [39]. Our results are consistent with these findings. However, unlike the previous studies that used national averages or non-spatial modeling, our approach uncovered regional differences in $NO_2$'s importance, suggesting local conditions may amplify its effects.

Previous studies provide contrasting conclusions about $O_3$'s relationship with AD dementia. In our study, $O_3$ emerged as an influential predictor, diverging from some previous literature that found either weak or no associations. For instance, Meo et al (2024) found no relationship between ground-level ozone and decreased global cognitive functions [35, 38]. A cohort study in London by Carey et al. (2018) provided similar results, specifying no positive exposure response between dementia and $O_3$ [40]. Contrastingly, a meta-analysis by Fu et al. (2020) provided positive evidence for the influence of $O_3$ on the development of AD dementia [41]. In addition, a population-based cohort study by Jung et al. (2015) reported an increased risk of AD due to exposure to higher levels of $O_3$ [42]. Given that $O_3$ levels tend to be higher in rural and suburban areas, this could contribute to the spatial associations observed [30, 31].

$PM_{2.5}$ has been established as a major contributing factor to cognitive decline, and our findings support its relevance, although it ranked slightly lower than $PM_{10}$, $NO_2$, and $O_3$. A cohort-based study by Yang et al. (2022) found that increased exposure to $PM_{2.5}$ had a positive association as a risk factor for AD in Zhejiang province, China [13]. Kioumourtzoglou et al. (2015) found significant positive associations between long-term $PM_{2.5}$ city-wide exposure and first hospital admission for AD among elderly populations in the Northeast US [43]. Additionally, a review by Shou et al. (2019) suggested that many particulate components of $PM_{2.5}$ can increase the risk of neurodegenerative diseases such as AD [35]. Although our results are generally consistent with previous studies, $PM_{2.5}$ did not rank as top predictor in the GWRF model, which highlights a potential discrepancy. This may be because, in certain areas, $PM_{10}$ or $NO_2$ are more closely linked to sources contributing to higher AD dementia prevalence. Additionally, the presence of multiple pollutants and spatial correlations may have reduced the apparent impact of $PM_{2.5}$ in our model.

CO and $SO_2$ emerged as the least impactful predictors of AD dementia prevalence in our study, consistent with the findings by Fu et al. (2020) [41]. A retrospective, population-based study in Taiwan by Chang et al. (2014) found that exposure to CO was associated with increased dementia risk [44]. Lin et al. (2021) conducted a case-control and city-by-city study comparing the progression of AD patients in cities with different pollutant levels and found that higher levels of both CO and $SO_2$ were associated with increased risk of AD cognitive deterioration [45]. Additionally, Meo et al. (2024) found that $SO_2$ had an association with a decrease in global cognitive functions, which counters $SO_2$ as our predictor with the lowest importance [42]. The discrepancies in our findings compared to previous studies may stem from differences in methodological approaches.

One limitation of the AD dementia prevalence data from Dhana et al. [14] is that rates were adjusted for age, sex, race/ethnicity, and education, but excluded smaller racial/ethnic groups such as Asian Americans and American Indian or Alaska Natives, potentially skewing estimates in affected regions. The lack of external validation is another limitation. Due to the absence of independent, nationally representative datasets, we relied on internal validation using a 15% test split, along with sensitivity analyses and model comparisons. County-level analyses may also obscure within-county variability, introducing ecological fallacy. The absence of sub-county prevalence data further limits spatial resolution, however, our study reflects the most granular analysis possible across the US.

Future research could benefit from incorporating higher-resolution data, such as census tract or ZIP code–level environmental and AD dementia data. This would allow for more granular assessments of spatial heterogeneity in AD dementia prevalence and air pollution exposure for more targeted interventions. We also aim to explore environmental toxins—pesticides, heavy metals (e.g., lead, mercury), and industrial chemicals—due to emerging links to neurodegeneration. Though large-scale genetic data are limited, proxies like family history and multi-level models may help capture interactions with comorbidities, behaviors, and social factors. Advanced ML techniques, especially ensemble methods, may improve predictive performance beyond individual models. Future studies should also explore spatial Bayesian hierarchical models, which explicitly account for spatial autocorrelation, provide more stable estimates, and support prior knowledge and uncertainty quantification—crucial for public health decisions.

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
