# OpenReview forum: "Long-term Exposure to Air Pollutants and Alzheimer’s Disease Dementia Prevalence Across the Contiguous United States: An Explainable Machine Learning Analysis"
_IEEE.org/EMBS/BHI/2025/Conference — BHI 2025_

### Official Review · Reviewer_G4Q9 · 2025-07-02
**Long-term exposure review**

**Confidence:** 3
**Clarity Of Writing:** excellent
**Clinical Significance:** excellent
**Methodological Novelty:** good
**Overall Rating:** 8
**Final Rating:** 8

**Experiments And Results:**

great

**Questions For The Authors:**

I am a little unsure about the format of the data. Is it a measurement of air pollutants for each county? Does that mean your N=the # of counties in the United States? Or do you have more than one data point per county? If this is true for the N value, did you randomly assign counties into training, testing, and validation? Or did you assign counties into the different splits proportionally by region?

**Strengths:**

-The purpose of the study is well outlined with many references to prior work that establish a strong foundation for the study

-The choice of models is well reasoned, particularly with how these models (local XGBoost and GWRF) address the geographical challenges of pollution models. Strong comparisons to global models also help strengthen the study.

-Many different metrics were used to evaluate each model, including multiple accuracy and interpretability measures.

-Good explanation of interactions between variables, and decision to not include interaction terms.

-Very clear figures that explain the results of the study very well. I particularly like Figure 2, showing how the primary pollutants change by region.

-Discussion provides context for every result, grounding the study in specific regional differences or key events (like wildfires).

**Summary Of The Paper:**

This study investigated how well specific air pollutants related to the prevalence of Alzheimer’s and dementia across the US. Comparing both local and global models, they showed that incorporating local context increases the performance of predictive environmental health models. They also show that PM10 is a prominent pollutant linked to AD in many counties in the US.

**Weaknesses:**

-It would be helpful if the model hyperparameters were placed in a table instead of being written in the text. This decreases readability.

-“Preliminary statistics show that the AD dementia prevalence ranged from 5.6% in Loving, Texas to 18.4% in Presidio, Texas.” – This only provides the context for a single region of the united states, what about the other regions?

---

### Official Review · Reviewer_vg2F · 2025-07-02
**Solid Environmental Health Study with Methodological Rigor Despite Some Limitations**

**Confidence:** 4
**Clarity Of Writing:** excellent
**Clinical Significance:** great
**Methodological Novelty:** great
**Overall Rating:** 7
**Final Rating:** 7

**Experiments And Results:**

good

**Questions For The Authors:**

The fundamental concern regarding this analysis centers on the validity of using Alzheimer's disease prevalence estimates derived from a single cohort study (CHAP) and applying them universally across all US counties. How do the authors address the potential bias introduced by this approach, particularly given that the Chicago-based model may not adequately capture regional variations in genetic predisposition, lifestyle factors, healthcare access, and other determinants that could significantly influence dementia prevalence patterns across diverse geographic and demographic contexts? Additionally, what sensitivity analyses were conducted to assess how uncertainties in the prevalence estimates might affect the conclusions about air pollutant relationships?

The temporal disconnect between the health data collection period (1993-2012) and the application to 2020 population estimates raises questions about the validity of assuming stable disease patterns over nearly three decades. During this period, significant changes in air quality regulations, healthcare practices, diagnostic criteria, and population demographics have occurred—how do the authors account for these temporal changes, and what evidence supports the assumption that relationships observed in the 1990s-2000s remain valid for contemporary populations? Furthermore, given that air pollution levels have generally improved over this timeframe due to regulatory interventions, how might this affect the interpretation of the dose-response relationships identified in the study?
The spatial modeling approach, while sophisticated, raises several technical questions about the robustness of the local relationships identified. How sensitive are the results to the choice of spatial bandwidth (k=92 neighbors) used in the GWRF model, and was this parameter systematically optimized or chosen arbitrarily? Given that the study identified substantial regional variations in model fit (R² ranging from <0.1 to 0.7), what specific factors might explain the poor model performance in certain regions, and how does this affect the reliability of conclusions drawn about air pollutant effects in these areas?

Finally, the ecological study design inherently limits causal inference, particularly when important confounders may vary spatially in ways that correlate with both air pollution exposure and Alzheimer's disease prevalence. How do the authors address the potential for unmeasured spatial confounding, particularly regarding factors such as urban versus rural lifestyle differences, healthcare infrastructure, genetic population structure, and socioeconomic gradients that might explain both pollution patterns and disease prevalence? What additional validation approaches, such as comparison with independent cohort studies or analysis of temporal trends, could strengthen confidence in the causal interpretation of these associations?

**Strengths:**

This study addresses an important public health question with substantial clinical and policy implications, providing valuable insights into environmental determinants of Alzheimer's disease across a large geographic scale. The methodological approach is comprehensive and well-designed, employing both global and local machine learning models to capture spatial heterogeneity that traditional approaches might miss. The use of GWRF represents a significant advancement over standard random forest models by accounting for spatial non-stationarity, which is particularly important for environmental health research where relationships can vary dramatically by geography. The long-term exposure assessment using 22 years of data (1999-2020) provides a robust foundation for examining chronic effects, and the inclusion of six different pollutants allows for comprehensive assessment of air quality impacts. The model evaluation is thorough, incorporating multiple performance metrics (R², RMSE, AIC) and proper validation procedures with appropriate train-test splits. The interpretability analysis using both Gini importance and SHAP values enhances understanding of pollutant contributions, while the spatial diagnostic tools (Local Moran's I, local R² values) provide valuable insights into model performance variations across regions.

**Summary Of The Paper:**

This study examines the relationship between long-term exposure to six air pollutants (PM2.5, PM10, NO2, CO, O3, SO2) and Alzheimer's disease dementia prevalence across US counties using machine learning approaches. The authors compare global models (Random Forest, XGBoost) with local spatial models (GWRF, Local XGBoost) using 1999-2020 pollution data. The GWRF model achieved the best performance (R²=54.38%), with PM10 emerging as the most influential predictor, followed by NO2, O3, and PM2.5. The study reveals substantial geographic variation in pollutant importance, with PM10 being the primary variable in 25.31% of counties.

**Weaknesses:**

The most significant limitation stems from the Alzheimer's disease prevalence data, which is derived from a single cohort study (Chicago Health and Aging Project) and extrapolated to all US counties, potentially introducing substantial bias and limiting the ability to capture true regional variations in disease risk. This approach assumes that demographic-adjusted prevalence patterns from one urban area apply universally across diverse geographic and cultural contexts, which may not be valid. The temporal mismatch between the prevalence model (based on 1993-2012 data) and the 2020 population estimates creates additional uncertainty. The exclusion of smaller racial/ethnic groups from the prevalence model may lead to biased estimates in areas where these populations are prevalent, further limiting the generalizability of findings. While the study focuses exclusively on air pollutants, it omits other important environmental and social determinants that could confound the relationships, such as healthcare access, socioeconomic factors, lifestyle variables, and genetic predisposition patterns that may vary geographically. The county-level analysis, while comprehensive, may mask important within-county variations in both exposure and disease prevalence, potentially leading to ecological fallacy issues. Additionally, the study lacks validation against independent datasets or external cohorts, making it difficult to assess the true generalizability of the findings beyond the specific modeling framework employed.

---

### Official Review · Reviewer_XNWm · 2025-07-11
**Spatial Modeling Without Causal Justification**

**Confidence:** 4
**Clarity Of Writing:** great
**Clinical Significance:** good
**Methodological Novelty:** fair
**Overall Rating:** 4
**Final Rating:** 7

**Experiments And Results:**

fair

**Questions For The Authors:**

1. Could the authors clarify why PM10 emerged as the most important pollutant in their analysis, when several prior studies (e.g., Mortamais et al., 2021; Zhang et al., 2023; Mollalo et al., 2025) have identified PM2.5 as more strongly associated with cognitive decline and Alzheimer’s disease?
It would be helpful to understand whether this difference is due to model structure, data characteristics, or other factors specific to the study context.

2. How sensitive are the findings to the choice of spatial modeling parameters—specifically the neighborhood size (k=92) and kernel type?
The paper briefly mentions these settings, but does not explain why they were selected or whether different values were tested. More detail on this would help clarify how these choices influence the results.


Mortamais, M., Gutierrez, L. A., de Hoogh, K., Chen, J., Vienneau, D., Carrière, I., ... & Berr, C. (2021). Long-term exposure to ambient air pollution and risk of dementia: Results of the prospective Three-City Study. Environment international, 148, 106376.
Zhang B, Weuve J, Langa KM, et al. Comparison of Particulate Air Pollution From Different Emission Sources and Incident Dementia in the US. JAMA Intern Med. 2023;183(10):1080–1089. doi:10.1001/jamainternmed.2023.3300
Mollalo A, Grekousis G, Florez H, Neelon B, Lenert LA, Alekseyenko AV. Alzheimer's Disease Dementia Prevalence in the United States: A County-Level Spatial Machine Learning Analysis. Am J Alzheimers Dis Other Demen. 2025 Jan-Dec;40:15333175251335570. doi: 10.1177/15333175251335570. Epub 2025 Apr 21. PMID: 40257111; PMCID: PMC12035167.
The last one also does a similar analysis at the county level.

**Strengths:**

The study effectively applies spatial machine learning models to capture geographic variation in the association between air pollution and Alzheimer’s disease prevalence. It employs long-term air pollution exposure data from 1999 to 2020, offering robust temporal depth. By comparing local and global ML models, the analysis highlights important spatial heterogeneity across counties. Additionally, the study provides clear visualizations and presents results at the county level, enhancing the granularity of its findings.

**Summary Of The Paper:**

The paper investigates the association between long-term exposure to ambient air pollutants and Alzheimer’s disease (AD) dementia prevalence across U.S. counties. Using machine learning models including Random Forest, XGBoost, and Geographically Weighted Random Forest (GWRF), the authors model spatial variation and identify the most influential pollutants. The study finds that PM10 is the top-ranked predictor of AD prevalence and presents regional variation in model performance. It suggests that place-based interventions targeting PM10 may help reduce AD burden.

**Weaknesses:**

The machine learning methods used—Random Forest, XGBoost, and GWRF—are fundamentally correlational tools developed for prediction, not for causal inference or policy intervention. While the study presents valuable spatial insights, it lacks a causal framework. No effort is made to address confounding, selection bias, or counterfactual reasoning. As such, ML models alone are not sufficient for drawing conclusions about intervention or informing policy recommendations.
The authors make causal-sounding claims throughout the paper that are not supported by the methodology. For example, statements such as:
•	“This would allow for more granular assessments of spatial heterogeneity… for more targeted interventions,”
•	“...underscoring the potential effectiveness of localized air quality interventions,” and
•	“...helping to inform more targeted and refined public health policies,”
are not methodologically justified and cannot be derived from purely correlational models.

---

### Official Review · Reviewer_bmFD · 2025-07-21
**Long-term Exposure to Air Pollutants and Alzheimer's Disease Dementia Prevalence Across the Contiguous United States: An Explainable Machine Learning Analysis**

**Confidence:** 3
**Clarity Of Writing:** good
**Clinical Significance:** good
**Methodological Novelty:** good
**Overall Rating:** 7

**Experiments And Results:**

good

**Questions For The Authors:**

1. Are there other environmental or demographic variables you plan to explore in future models to better capture the complexity of spatial variation in AD dementia prevalence?
2. You noted discrepancies between your findings and previous literature for pollutants. How do you plan to investigate these discrepancies further?
3. What specific public health interventions or policy recommendations would you propose, particularly for urban areas, to mitigate dementia related to PM10 exposure?
4. Have you considered how the interplay between air pollution and other established risk factors for dementia (e.g., genetic predispositions, lifestyle factors) might be incorporated into future spatial machine learning analyses?

**Strengths:**

The study compiled long-term data for six air pollutants, offering a robust dataset for analysis.
The researchers analyze two machine learning models, global Random Forest, global XGBoost, geographically weighted random forest (GWRF), and local XGBoost, to examine the relationship between air pollutants and AD dementia prevalence.
The study effectively addresses the limitation of global models by employing local models like GWRF and Local XGBoost, which account for geographical variations in environmental exposures and health impacts.
The use of Gini feature importance and SHAP values allowed for a thorough assessment of the relative contribution of each pollutant and clear interpretation of model outputs.
The authors acknowledge that discrepancies with previous studies may stem from differences in methodological approaches, demonstrating a critical perspective on their findings.

**Summary Of The Paper:**

This study investigates the long-term impact of air pollutant exposure on Alzheimer's Disease (AD) dementia prevalence across the contiguous United States, employing global and local machine learning models.

**Weaknesses:**

The regression model for prevalence estimation was based solely on the Chicago Health and Aging Project and applied across all U.S. counties, which may limit the ability to observe regional variations in Alzheimer's risk.
Smaller racial/ethnic groups were excluded from the model, potentially skewing AD dementia prevalence estimates in areas with higher proportions of these groups.